# Translation, Cross-Cultural Adaptation, and Psychometric Validation of the Authentic Nurse Leadership Questionnaire for the Portuguese Context: A Methodological Study

**DOI:** 10.3390/nursrep15100362

**Published:** 2025-10-09

**Authors:** Pedro Lucas, Ana Gaspar, Paulo Cruchinho, Mafalda Inácio, Ana Rita Figueiredo, Luísa Dias, Paula Agostinho, João Oliveira, Marie Giordano-Mulligan, Elisabete Nunes, Patrícia Costa

**Affiliations:** 1Nursing Research Innovation and Development Centre of Lisbon (CIDNUR), Nursing School of Lisbon, 1600-190 Lisbon, Portugal; prlucas@esel.pt (P.L.); ana-gaspar@campus.esel.pt (A.G.); pjcruchinho@esel.pt (P.C.); mafaldainacio@esel.pt (M.I.); l.dias@esel.pt (L.D.); paulamtragostinho@sapo.pt (P.A.); joao.oliveira2@ulssjose.min-saude.pt (J.O.); enunes@esel.pt (E.N.); patriciacosta@esel.pt (P.C.); 2Nursing Administration Department, Nursing School of Lisbon, Avenida Prof. Egas Moniz, 1600-190 Lisbon, Portugal; 3Unidade Local de Saúde de São José, Rua José António Serrano, 1150-199 Lisboa, Portugal; 4Huntington Hospital, 270 Park Avenue, Huntington, NY 11743, USA; mmulligan5@northwell.edu

**Keywords:** authentic leadership, psychometric properties, validation studies, nursing administration research

## Abstract

**Background**: Authentic leadership is characterized by the authenticity and self-awareness of the leader, who acts with transparency and promotes positive outcomes in clinical practice and team management. In Portugal, there isn’t a tool available to assess nurses’ perceptions of authentic leadership in nursing. This study aimed to translate and cross-culturally adapt the Authentic Nurse Leadership Questionnaire (ANLQ) for the Portuguese context and to evaluate its psychometric properties. This instrument assesses nurses’ perceptions of the authentic leadership exercised by their leader. **Methods**: A methodological, descriptive, cross-sectional study with a quantitative approach was conducted. The translation and cross-cultural adaptation process followed the recommendations of internationally recognized guidelines. The Authentic Nurse Leadership Scale—Portuguese version (ANLS-PT) was administered to a sample of 406 nurses from various functional units in three primary healthcare centers. Exploratory and confirmatory factor analysis techniques were used. Reliability was established through a test–retest administration to 22 nurses at two different times, with a two-week interval. The internal consistency of the scale was assessed using Cronbach’s Alpha. **Results**: An instrument with 29 items and 3 dimensions was obtained, explaining 68.3% of the total variance. The identified dimensions were Caring and Decision-Making, Self-Awareness, and Relational Integrity and Ethics. The overall instrument showed an internal consistency of 0.97. **Conclusions**: The ANLS-PT proved to be a valid, reliable, and robust tool for assessing authentic leadership in the Portuguese cultural context and can be used in various nursing practice contexts.

## 1. Introduction

The proposal of authentic leadership emerges from the need to explore how relationships between leaders and followers develop, and to understand the leader through the principles of Positive Psychology and Humanistic Psychology [1]. The concept of authentic leadership was developed by Luthans and Avolio [2] as “a process based on positive psychological capabilities and a highly developed organizational context, which results in greater self-awareness and self-regulated positive behaviors on the part of leaders and employees, fostering positive self-development” (p. 243).

Authentic leaders achieve a high level of authenticity because they know who they are, what they believe in, and what they value. They act in accordance with these beliefs and values, interacting transparently with employees [3]. The defining attributes of authentic leadership are as follows: (a) self-awareness—the leader knows their strengths, weaknesses, beliefs, and emotions, using this self-knowledge to guide their actions, it is the basis of authentic leadership; (b) transparency—the leader communicates honestly, sharing their thoughts and feelings with employees, creating relationships based on trust and openness; (c) balanced processing—the leader seeks to analyze information fairly and thoughtfully, considering various perspectives without personal distortions; (d) internal moral perspective—the leader maintains a strong internal ethical sense, aligning personal values with their decisions and actions at work; (e) caring—the leader demonstrates concern for the well-being of employees, creating a healthy, cohesive and supportive work environment; (f) shared decision-making—the leader involves and values the contributions of employees; (g) moral/ethical courage—the leader maintains ethical values even in the face of adversity [4,5,6,7,8,9,10]. Generally, leadership is a competence that can be developed [5,9,11,12], associated with behaviors and practices throughout professional experience, and also during undergraduate and postgraduate nursing education [13].

Leadership is one of the conceptual dimensions of the nursing practice environment and is associated with the quality and safety of care [12,14,15].

In nursing, authentic leadership has received special attention due to its impact on clinical practice and team management [9,16]. Studies reveal that authentic leadership promotes the following: higher levels of satisfaction, engagement, and organizational commitment [17,18,19,20]; employee well-being and satisfaction, a positive work environment, and improved performance and safety [4]; and enhanced performance of nursing teams regarding patient comfort and ethical orientation in nurses [21].

In Portugal, there is no instrument available to assess nurses’ perceptions of authentic leadership in nursing. Given the importance and relevance of this phenomenon, it was considered essential to validate and adapt the Authentic Nurse Leadership Questionnaire (ANLQ), developed by Giordano-Mulligan [22,23], for the Portuguese population. This instrument aims to evaluate nurses’ perceptions of their leaders’ authentic leadership. Thus, the objective of this study was to translate, cross-culturally adapt, and validate the ANLQ, resulting in its Portuguese version.

## 2. Materials and Methods

### 2.1. Design

A methodological, cross-sectional study with a quantitative approach was designed [24,25] to address the following research question: “Does the ANLQ demonstrate good psychometric properties for the Portuguese population?”

The Guidelines for Reporting Reliability and Agreement Studies (GRRAS) [26] were followed.

### 2.2. Target Population, Sample, and Sampling Technique

The target population for this study was 510 nurses from three primary healthcare centers in Lisbon and Tagus Valley Regional Health Administration.

The inclusion criteria for the study were nurses working in healthcare organizations in Portugal, with more than two years of professional experience, who gave their permission to participate in the questionnaire.

A sample of 406 nurses was obtained, selected through a non-probabilistic convenience sampling technique [24,25].

### 2.3. Data Collection Instrument

The instrument used was the ANLQ, developed by Giordano-Mulligan [22,23]. The ANLQ aims to assess nurses’ perception of authentic leadership exercised by their immediate leaders, being specifically designed for the nursing context. It consists of a total of 29 items organized on a five-point Likert-type frequency scale (0 = never; 4 = always). These items evaluate five distinct dimensions: self-awareness; moral ethical courage; relational integrity; shared decision-making; and caring. These dimensions were defined based on a conceptual model anchored in the disciplinary values of nursing, namely personal integrity, transparency, and altruism, demonstrating robust evidence of validity and reliability in the exploratory and confirmatory factor analyses conducted during the instrument’s development process [22,23] and obtained an internal consistency value of 0.99 [22,23]. The reliability values of the ANLQ test–retest ranged from 0.89 to 0.99 [22].

### 2.4. Translation and Cross-Cultural Adaptation Process

The process of translation and cross-cultural adaptation of the ANLQ followed the methodological guidelines proposed by Beaton et al. [27,28], Cruchinho et al. [29], and Sousa and Rojjanasrirat [30].

The first stage involved the initial translation of the original English version into Portuguese. This translation was performed independently by two bilingual and bicultural translators. One translator had prior knowledge of the instrument’s subject area and familiarity with the study’s objectives, which ensured greater cultural and idiomatic appropriateness of the translation (T1). The second translator, on the other hand, had no prior knowledge of the topic or the instrument’s objectives (T2), which ensured a more literal translation, free from interpretive biases.

In the second stage, an expert panel meeting was held. During this meeting, the T1 and T2 versions were compared, discussed, and harmonized, resulting in a consensus version (synthesis version) that constituted the first Portuguese version of the instrument.

The third stage, according to Beaton et al. [27,28], Cruchinho et al. [29], and Sousa and Rojjanasrirat [30], was the back-translation of the synthesis version into the original language, English. This procedure was conducted by two independent translators, who were also bilingual and bicultural but without training in the health field (BT1 and BT2). The objective of this stage was to evaluate the conceptual and semantic equivalence between the versions, to ensure that the translated content accurately reflected the original meanings of the instrument’s items [27,28,29,30].

In the fourth stage, an expert committee was established to critically analyze the entire adaptation process. This committee included a professor with a PhD in linguistics, two professors with PhDs in nursing with consolidated experience in nursing leadership, and two nurses with a master’s degree with methodological knowledge in translation and cross-cultural adaptation processes. The experts conducted a comparative analysis of the back-translated versions, the Portuguese synthesis version, and the original English version to ensure semantic, idiomatic, cultural, and conceptual equivalence [27,28,29,30]. It was found that versions T1 and T2 coincided entirely on 17 items, with the remaining differences classified as semantic variations of synonyms, with no impact on the meaning of the items. These differences were resolved by expert consensus.

In the fifth stage, the pre-final version of the instrument was pre-tested on a sample of 30 nurses from the target population. Each participant received the pre-final version of the instrument, an application guide for the scale, and a form for recording suggestions, comments, or observations. Before its application, the objectives of the pre-test were clarified, emphasizing that the instrument’s structure (number of items, internal organization, and response format) could not be modified. The pre-test aimed to evaluate the clarity, comprehension, and relevance of the items, as well as the cultural acceptability and adequacy of the expressions used. The response time ranged from five to fifteen minutes. As a result, no comprehension difficulties were identified by the participants, nor were any doubts or ambiguities reported regarding the content, wording, or format of the items, which demonstrated the instrument’s clarity and suitability. Therefore, it was concluded that no further reformulations were necessary.

Finally, to ensure greater methodological rigor, the final version of the instrument was shared with the original author of the scale, who reviewed and approved the adaptation. This concluded the translation and adaptation process, defining the final version of the instrument as the ANLS-PT.

### 2.5. Data Collection

Data collection took place between May and August 2023, using an online questionnaire (Google Forms^®^) structured in two parts:Part A: A sociodemographic and socio-professional questionnaire (gender, age, professional experience, academic qualifications, type of functional unit, professional category, management functions);Part B: The translated and adapted version of the ANLS.

### 2.6. Data Analysis

Statistical analysis, including exploratory factor analysis, was performed using IBM SPSS Statistics^®^, version 22.0. For the structural equation modeling, corresponding to the confirmatory factor analysis, the IBM^®^ SPSS^®^ AMOS^®^, version 22 (SPSS Inc., Armonk, NY, USA) software was used.

### 2.7. Ethical Considerations

This study was previously approved by the Health Ethics Committee of the Lisbon and Tagus Valley Regional Health Administration, under opinion number 131/CES/INV/2022, issued on 14 April 2023.

All participants were properly informed about the research objectives, the voluntary nature of their participation, and the confidentiality of the data collected. Written informed consent was obtained from all participants before data collection began. The study was conducted in full compliance with the ethical principles established in the Declaration of Helsinki [31].

## 3. Results

### 3.1. Sociodemographic Characteristics of the Sample

In the sample of 406 nurses who participated in the study, 94.7% were female. According to the National Institute of Statistics, in 2021, the percentage of female nurses in Portugal was 82.4% [32].

The age distribution of participants was mainly concentrated in the 45 to 49 years (18.9%) and 53 to 57 years (18%) age groups, with an average age of 49 years. This predominance of professionals over the age of 45 reflects an experienced professional group, with many nurses having increased responsibilities and a more consolidated view on leadership practices. This is relevant for the interpretation of the results, as the experience acquired and developed over the years can influence perceptions of leadership and, consequently, of authentic leadership.

This finding is further corroborated by the data obtained from professional experience, which showed that the majority (65.6%) of nurses had between 24 and 36 years of practice. Specifically, 34.5% had 24–29 years and 31.1% had 29–36 years, indicating a sample composed of professionals with well-established careers.

Academic qualifications showed a predominance of bachelor’s degrees (48.1%), followed by master’s degrees (31.6%) and postgraduate courses (19.4%). This finding is relevant as higher academic qualifications are often associated with enhanced critical thinking and the development of advanced management and decision-making skills, which may significantly influence the perception of authentic leadership attributes.

In terms of professional category, 51.5% of the nurses were general nurses, while 46.6% were specialist nurses (with 12% of these in management positions), and 1.9% were nurse managers.

Finally, concerning the distribution of the sample across Functional Units, 51.9% were from Family Health Units, 26.2% from Community Care Units, 19.9% from Personalized Health Care Units, and 1.9% from Public Health Units.

### 3.2. Psychometric Properties Analysis

#### 3.2.1. Reliability

The internal consistency of the Portuguese version of the ANLS was evaluated using Cronbach’s Alpha coefficient, which yielded a value of 0.97. To complete the reliability analysis, we calculated McDonald’s Omega and obtained a value of 0.98. According to Almeida [26], this reflects a very good internal consistency. Internal consistency per dimension was also high: “Caring and Decision-Making” (α = 0.98); “Self-Awareness” (α = 0.95); and “Relational Integrity and Ethics” (α = 0.85), indicating internal homogeneity of the items within each factor.

Furthermore, a test–retest procedure was conducted on a subgroup of 22 nurses to assess temporal reliability and stability, with a two-week interval between administrations in accordance with the recommendations of Cruchinho et al. [29].

We have included Table 1 with the item-total correlation below.

#### 3.2.2. Content Validity

Content validity was ensured during the initial stages of the translation and cross-cultural adaptation process, as recommended by Gray and Grove [24] and Souza et al. [33]. The statistical analysis of the items, standard deviation, and symmetry is shown in Table 2.

#### 3.2.3. Construct Validity

Construct validity was analyzed in two complementary phases: exploratory factor analysis and confirmatory factor analysis.

Exploratory factor analysis

The factorial structure of the ANLS-PT instrument was explored using the 29 items from the translated and adapted version. The analysis began by verifying the data’s suitability for factor analysis, which involved calculating the Kaiser–Meyer–Olkin (KMO) index and performing Bartlett’s test of sphericity, following the methodological recommendations of Almeida [34], Marôco [35], and Grove et al. [24]. A KMO value of 0.97 was obtained, indicating an excellent correlation among variables and high sample adequacy for factor extraction. Bartlett’s test of sphericity (7150.6) was significant (*p* < 0.001), confirming the presence of significant correlations between the items and reinforcing the relevance of an exploratory factor analysis.

Subsequently, a Varimax orthogonal rotation was applied. The following criteria were used for factor retention: eigenvalues greater than or equal to 1, a minimum of two items per factor, loadings of ≥0.40, and no significant cross-loadings. The internal consistency of each dimension was assessed using Cronbach’s Alpha coefficient, with values of ≥0.85 considered indicative of good reliability [34].

The exploratory factor analysis resulted in the extraction of three factors, which collectively explained 68.30% of the total variance. All 29 items were retained and loaded onto one of the factors:Factor 1—Caring and Decision-Making: included 16 items (items 10–12, 16–18, 20–29) and explained 39.63% of the total variance. This dimension assesses the nurse leader’s attributes related to the ability to make evidence-based and informed decisions, while integrating ethics, empathy, and genuine, altruistic care for oneself and others.Factor 2—Self-Awareness: Composed of 8 items (items 1–8), this factor explained 27.30% of the total variance. It encompasses aspects of the nurse leader’s intrinsic motivation, personal vision, resilience, self-confidence, and critical reflection, all of which relate to their self-awareness and the impact they have in the professional context.Factor 3—Relational and Ethical Integrity: This factor grouped together 5 items (items 8, 13–15, 19) and explained 8.67% of the total variance. This dimension reflects the nurse leader’s ability to establish trusting relationships, characterized by active listening, honesty, respect, and the coherence between personal values and actions, even under pressure.

It was the first approach in EFA that resulted in the three factors, insofar as the indices obtained were very good, namely the alphas of each factor.

Following the conclusion of the process, the final configuration of the Portuguese version of the instrument, now named ALNS-PT, was established (Table 3). 

The resulting three-factor structure was conceptually coherent with the theoretical assumptions of authentic leadership and the original constructs defined by Giordano-Mulligan [22,23]. The high explained variance and robust reliability coefficients support the validity of the model and the instrument’s adequacy for the Portuguese cultural context.

Confirmatory factor analysis

The three-factor structure of the ANLS-PT was tested using confirmatory factor analysis with the IBM^®^ SPSS^®^ AMOS^®^, version 22 software. The evaluation of univariate and multivariate normality was based on the skewness (Sk) and kurtosis (Ku) coefficients, with no serious violations of statistical normality being observed (|Sk| < 3; |Ku| < 10). The identification of multivariate outliers was performed using Mahalanobis distance (D^2^) com 0.01.

The quality of the overall model fit was analyzed using multiple fit indices. The local fit was assessed based on the modification indices provided by the program, as well as on theoretical assumptions (values > 11; *p* < 0.001) [36]. A total of 54 covariance corrections were made, which resulted in an improvement in the overall fit indices, with the model achieving a very good fit: PCMIN/DF = 1.689, RMR = 0.024, GFI = 0.855, AGFI = 0.803, Delta2/NNFI = 0.969, CFI = 0.969, RMSEA = 0.058, MECVI = 3.910, and SRMR = 0.0268.

Furthermore, all items showed significant standardized factorial weights (λ > 0.50), indicating a substantial contribution to their respective factors.

No specific modifications were made because there was no need, and we obtained very good final model fit indices.

Figure 1 presents the final ANLS-PT model with the standardized factorial weights and the corrected covariances. The resulting configuration reflects a stable factorial structure, which confirms the robustness of the proposed model for application in the Portuguese nursing population.

## 4. Discussion

The translation and cross-cultural adaptation of the ANLS for the Portuguese context was performed following the methodological recommendations of Beaton et al. [27,28], and Sousa and Rojjanasrirat [30]. The objective was to ensure conceptual, semantic, and idiomatic equivalence with the original version developed by Giordano-Mulligan [22,23]. This methodological process followed a systematic and rigorous approach to ensure not only the fidelity of the translation but also the cultural and linguistic relevance of the items for the target population. The sociocultural specificities of the Portuguese context were considered, which contributed to the clarity, relevance and comprehension of the items, thus ensuring the content validity of the adapted version. As a result, the Portuguese version of the instrument proved to be conceptually equivalent to the original, with all items maintaining their integrity and relevance to the construct being measured.

The analysis of the instrument’s psychometric properties was conducted through a methodologically rigorous process involving a sample of 406 nurses working in primary healthcare. The initial phase consisted of an internal consistency evaluation, which yielded a Cronbach’s Alpha coefficient of 0.97, indicating excellent reliability according to Almeida [34]. This value aligns with the data from the original instrument, developed by Giordano-Mulligan [22,23], which reported a coefficient of 0.99. This is also consistent with the findings of Wang et al. [37], who reported a value of 0.97 for their version adapted to the Chinese cultural context.

The construct’s validity was assessed through an exploratory factor analysis, followed by a confirmatory factor analysis, in accordance with best methodological practices recommended for psychometric validation studies [27,28,30]. This sequential approach allowed us not only to explore the latent structure of the instrument, but also to empirically test the adequacy of the identified model to the sample under study—Portuguese nurses.

The exploratory factor analysis revealed a structure composed of three factors—Caring and Decision-Making, Self-Awareness, and Relational Integrity and Ethics—which collectively explained 68.3% of the total variance. This structure demonstrated conceptual coherence and showed general alignment with the theoretical assumptions underlying the original scale. However, it is important to note that the configuration obtained diverged from the factorial structure identified in both the scale’s initial development study by Giordano-Mulligan [22,23] and the subsequent translation and cross-cultural adaptation by Wang et al. [37] for the Chinese cultural context.

The original version of the instrument, developed by Giordano-Mulligan [22,23], is based on a robust theoretical model grounded in Jean Watson’s Theory of Caring and the principles of authentic leadership proposed by Avolio and Gardner in 2005 [38]. This model includes three central conceptual domains—Personal Integrity, Transparency, and Altruism—that support the five identified dimensions: Self-Awareness, Moral Ethical Courage, Relational Integrity, Shared Decision-Making, and Caring, with a total explained variance of 76.2%. Similarly, Wang et al. [37] confirmed this five-factor structure in their study, reporting a total explained variance of 82.6%. Despite the Portuguese version showing a structural reorganization that reduced the number of dimensions to three (Table 4), the semantic content and theoretical principles underpinning the five original dimensions were preserved.

Regarding the confirmatory factor analysis, the results demonstrated the stability of the three-factor structure of the Portuguese ANLS version, showing a very good overall fit to the study population with the following values: PCMIN/DF = 1.689, RMR = 0.024, GFI = 0.855, AGFI = 0.803, Delta2/NNFI = 0.969, CFI = 0.969, RMSEA = 0.058, MECVI = 3.910, and SRMR = 0.0268. These indices are within the parameters recommended by Marôco [36], providing solid empirical support for the validity of the three-dimensional model identified in the exploratory factor analysis and confirming the adequacy of the proposed structure for the Portuguese cultural context.

In the original study, Giordano-Mulligan [22] tested two confirmatory factor analysis models: a three-factor model and a five-factor model. Both demonstrated acceptable fit quality, with the five-factor model obtaining the best fit indices (RMSEA = 0.08, IFI = 0.93, and PNFI = 0.76). Similarly, Wang et al. [37], in their Chinese adaptation of the ANLS, also validated the five-factor model proposed by the original study, reporting values considered adequate: X2/df = 1.104, GFI = 0.905, AGFI = 0.887, RMSEA = 0.020, NFI = 0.915, TLI = 0.990, CFI = 0.991, and PGFI = 0.756.

The divergence between the Portuguese model (three factors) and the original and Chinese models (five factors) may reflect cultural and contextual variations in how nurses perceive the attributes of authentic leadership. The reconfiguration of the factorial model in the Portuguese study resulted in three consolidated dimensions—Caring and Decision-Making, Self-Awareness, and Relational Integrity and Ethics—which integrate the core concepts of the original dimensions and were semantically adapted to the Portuguese sociocultural reality. This phenomenon is consistent with the literature on cross-cultural adaptation, which acknowledges the possibility of factorial restructuring during the validation process without compromising the instrument’s validity [30,33].

Based on these findings, it is considered that the congruence between the observed data and the theoretical model confirms the robustness of the structure resulting from the cross-cultural adaptation process and establishes the Portuguese version as a reliable and valid instrument for nursing research and professional practice.

### 4.1. Implications for Practice

The Portuguese version of the ANLS, is considered a relevant and innovative contribution to nursing management and research. By demonstrating robust psychometric properties and ensuring conceptual coherence with the theoretical foundations of authentic leadership, this instrument allows for a rigorous and contextualized assessment of nurses’ perceptions of leadership behaviours aligned with the values of care, relational ethics and shared decision-making.

In the context of nursing management, the systematic use of this scale will enable leaders to engage in critical reflection on their practices, identifying strengths and areas for improvement in the dimensions evaluated: Caring and Decision-Making, Self-Awareness, and Relational Integrity and Ethics. This self-assessment and reflection, supported by objective data, will promote alignment between individual values and organisational principles, thus contributing to the construction of more effective individual development plans.

From a research perspective, the validation of the Portuguese version of the instrument opens new possibilities for further exploration of the phenomenon of authentic leadership in the national context. The availability of a culturally adapted measure will not only enable descriptive and comparative studies between different regions, institutions and organisational cultures, but also explore associations between leadership and the nursing practice environment, quality and safety of care provided, job satisfaction, motivation, organisational commitment, among other organisational behaviour variables.

### 4.2. Limitations

One of the main limitations of this study is the absence of previously validated reference instruments for the Portuguese context, which prevented the establishment of external correlations. This gap hindered the assessment of the ANLS-PT’s criterion validity, restricting data triangulation with convergent measures. This limitation stems from the scarcity of specific, validated instruments in Portugal that address constructs conceptually related to authentic leadership.

Additionally, we consider that the cross-sectional design of the study did not allow for the evaluation of the scale’s temporal stability or its sensitivity to changes resulting from training interventions or professional experiences. Therefore, conducting longitudinal studies emerges as a key opportunity for future research. Such studies would allow for the assessment of the temporal consistency of scores and the instrument’s ability to detect variations over time.

According to some perspectives, two weeks between the test and retest may be too short a period between data collection. This period may constitute a limitation of the study.

The fact that subjects were recruited from three primary healthcare centers using a non-probabilistic sampling method may have introduced potential sampling bias.

## 5. Conclusions

The results of this study validate the ANLS-PT as a psychometrically robust and culturally appropriate instrument for assessing Authentic Leadership in Nursing in Portugal. The translation and cross-cultural validation of the scale ensure that the ANLS-PT accurately reflects the specific cultural and professional nuances of the Portuguese context, allowing for a rigorous evaluation of nurses’ perceptions of authentic leadership.

The implications for both research and healthcare delivery are significant. The ANLS-PT offers the scientific community a validated and reliable data collection tool, adapted to the Portuguese cultural context, which will contribute to the development of comparative and longitudinal studies on nursing leadership. For nurse managers, this instrument will provide reliable results on nurses’ perceptions of their leadership competencies. This will enable the identification of areas for improvement and the implementation of strategies to strengthen leadership, enhance the quality of nursing care, and improve patient safety.

To strengthen the conclusions and ensure data representativeness, a nationwide expansion of the study is recommended, covering different care practice contexts and a larger number of healthcare institutions. Furthermore, applying the ANLS-PT in various regions of the country will allow for the evaluation of its validity in diverse cultural and organizational contexts, contributing to the instrument’s overall robustness.

## Figures and Tables

**Figure 1 nursrep-15-00362-f001:**
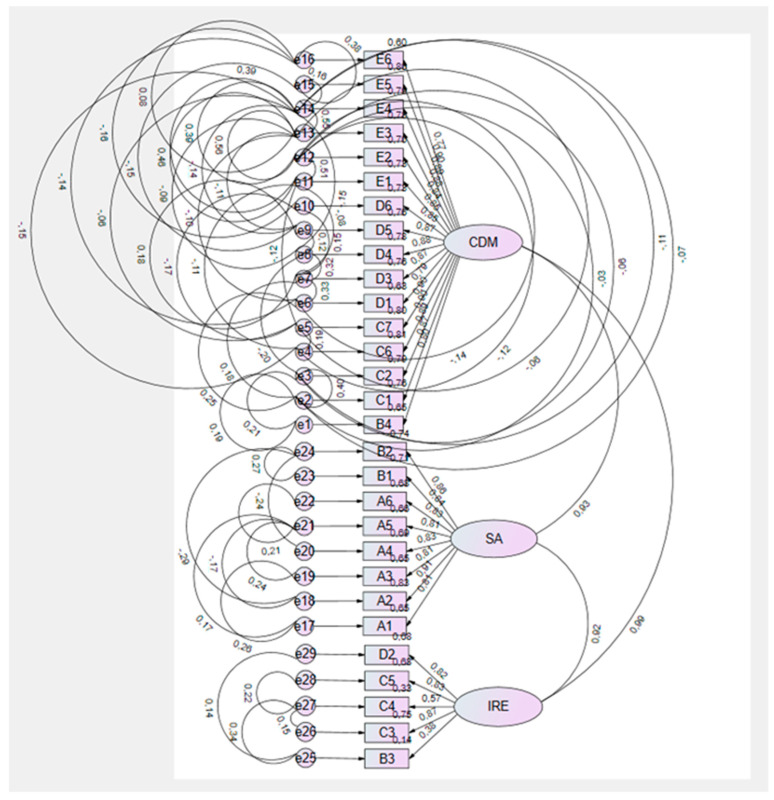
Three-factor model of the ANLS-PT.

**Table 1 nursrep-15-00362-t001:** The item-total correlation.

	2	3	4	5	6	7	8	9	10	11	12	13	14	15	16	17	18	19	20	21	22	23	24	25	26	27	28	29
**1**	0.759	0.742	0.643	0.714	0.704	0.698	0.681	0.306	0.612	0.675	0.661	0.664	0.450	0.607	0.642	0.648	0.544	0.574	0.635	0.631	0.637	0.577	0.630	0.611	0.634	0.626	0.662	0.647
**2**	1.000	0.740	0.771	0.703	0.743	0.748	0.715	0.338	0.699	0.751	0.766	0.701	0.494	0.658	0.713	0.721	0.668	0.647	0.764	0.722	0.704	0.760	0.759	0.734	0.784	0.761	0.770	0.647
**3**		1.000	0.650	0.735	0.664	0.700	0.702	0.301	0.582	0.659	0.668	0.687	0.490	0.630	0.653	0.667	0.574	0.601	0.614	0.664	0.638	0.584	0.629	0.636	0.686	0.649	0.686	0.625
**4**			1.000	0.688	0.715	0.675	0.633	0.268	0.642	0.674	0.684	0.608	0.369	0.585	0.686	0.667	0.619	0.563	0.670	0.666	0.690	0.692	0.672	0.661	0.678	0.676	0.693	0.587
**5**				1.000	0.745	0.675	0.667	0.310	0.617	0.665	0.702	0.705	0.427	0.634	0.669	0.715	0.554	0.633	0.634	0.634	0.692	0.623	0.592	0.612	0.642	0.658	0.689	0.630
**6**					1.000	0.697	0.703	0.235	0.703	0.709	0.737	0.650	0.416	0.621	0.692	0.679	0.548	0.628	0.668	0.665	0.664	0.614	0.611	0.623	0.672	0.631	0.669	0.581
**7**						1.000	0.808	0.294	0.721	0.732	0.726	0.731	0.428	0.632	0.677	0.741	0.613	0.662	0.676	0.712	0.703	0.629	0.638	0.679	0.708	0.687	0.736	0.640
**8**							1.000	0.387	0.728	0.731	0.750	0.786	0.461	0.665	0.693	0.763	0.639	0.674	0.674	0.703	0.719	0.650	0.676	0.678	0.715	0.703	0.700	0.633
**9**								1.000	0.350	0.362	0.310	0.329	0.475	0.315	0.312	0.359	0.307	0.389	0.313	0.326	0.326	0.326	0.299	0.338	0.308	0.317	0.316	0.250
**10**									1.000	0.750	0.772	0.714	0.443	0.606	0.707	0.708	0.598	0.641	0.700	0.732	0.711	0.672	0.700	0.691	0.722	0.707	0.708	0.589
**11**										1.000	0.859	0.754	0.521	0.702	0.796	0.769	0.747	0.746	0.799	0.780	0.759	0.755	0.735	0.705	0.758	0.741	0.768	0.676
**12**											1.000	0.811	0.534	0.739	0.803	0.785	0.667	0.692	0.760	0.745	0.755	0.760	0.751	0.731	0.798	0.765	0.811	0.699
**13**												1.000	0.565	0.737	0.792	0.812	0.633	0.696	0.695	0.729	0.746	0.717	0.673	0.676	0.734	0.725	0.742	0.618
**14**													1.000	0.583	0.546	0.520	0.427	0.476	0.480	0.498	0.407	0.478	0.462	0.482	0.512	0.510	0.485	0.427
**15**														1.000	0.755	0.763	0.643	0.673	0.700	0.723	0.690	0.656	0.704	0.690	0.721	0.716	0.727	0.691
**16**															1.000	0.842	0.732	0.770	0.796	0.806	0.781	0.783	0.724	0.695	0.762	0.739	0.793	0.691
**17**																1.000	0.713	0.744	0.790	0.805	0.796	0.734	0.725	0.732	0.766	0.765	0.772	0.654
**18**																	1.000	0.725	0.798	0.710	0.701	0.685	0.678	0.602	0.661	0.691	0.696	0.543
**19**																		1.000	0.806	0.766	0.789	0.652	0.655	0.638	0.693	0.682	0.697	0.619
**20**																			1.000	0.841	0.806	0.786	0.730	0.711	0.770	0.796	0.782	0.646
**21**																				1.000	0.803	0.732	0.763	0.735	0.762	0.755	0.787	0.666
**22**																					1.000	0.765	0.727	0.706	0.731	0.736	0.744	0.651
**23**																						1.000	0.743	0.728	0.756	0.765	0.782	0.615
**24**																							1.000	0.860	0.869	0.853	0.807	0.712
**25**																								1.000	0.884	0.846	0.793	0.680
**26**																									1.000	0.908	0.834	0.744
**27**																										1.000	0.847	0.704
**28**																											1.000	0.798

**Table 2 nursrep-15-00362-t002:** Statistical analysis of items, standard deviation and symmetry.

**Items**	**1**	**2**	**3**	**4**	**5**	**6**	**7**	**8**	**9**	**10**	**11**	**12**	**13**	**14**	**15**
Standard Deviation	0.811	0.989	0.891	0.993	0.884	0.829	0.888	0.928	0.876	1.013	1.015	0.937	0.961	1.020	0.893
Asymmetry	0.495	0.582	0.160	0.620	0.672	0.803	0.489	0.535	0.001	0.620	0.990	1.069	0.651	0.378	0.432
Kurtosis	0.233	0.304	0.811	0.451	0.182	0.125	0.767	0.407	0.402	0.226	0.428	0.812	0.259	0.150	0.138
**Items**	**16**	**17**	**18**	**19**	**20**	**21**	**22**	**23**	**24**	**25**	**26**	**27**	**28**	**29**	
Standard Deviation	1.040	0.948	0.976	0.924	1.005	0.993	0.898	0.878	1.059	0.936	0.931	0.991	0.990	0.911	
Asymmetry	0.771	0.668	0.489	0.526	0.638	0.782	0.800	0.969	0.691	0.369	0.579	0.657	0.914	0.619	
Kurtosis	0.073	0.152	0.308	0.286	0.029	0.263	0.316	0.449	0.030	0.394	0.146	0.182	0.382	0.216	

**Table 3 nursrep-15-00362-t003:** Final factor structure: Distribution of items by factor, factor loadings, total variance explained, and Cronbach’s Alpha coefficients.

Total Items	Items	Dimensions		
		Caring and Decision-Making	Self-Awareness	Relational Integrity and Ethics
16	10	0.592		
	11	0.680		
	12	0.670		
	16	0.730		
	17	0.685		
	18	0.728		
	20	0.795		
	21	0.749		
	22	0.711		
	23	0.760		
	24	0.807		
	25	0.754		
	26	0.971		
	27	0.803		
	28	0.759		
	29	0.622		
8	1		0.783	
	2		0.653	
	3		0.746	
	4		0.645	
	5		0.765	
	6		0.765	
	7		0.698	
	8		0.648	
	9			0.858
5	13			0.326
	14			0.718
	15			0.317
	19			0.330
Explained Variance		39.63	27.30	8.67
Cronbach’s Alpha		0.98	0.95	0.85
		0.97		

**Table 4 nursrep-15-00362-t004:** Comparison of the ANLS-PT with the original version by Giordano-Mulligan [22,23] and the version translated and adapted for the Chinese cultural context by Wang et al. [37].

Version	Domains/Dimensions		No. of Items	Items	Cronbach’s Alpha	
ANLS-PT	Caring and Decision-Making		16	10, 11, 12, 16, 17, 18, 20, 21, 22, 23, 24, 25, 26, 27, 28, 29	0.98	0.97
	Self-Awareness		8	1, 2, 3, 4, 5, 6, 7, 8	0.95	
	Relational Integrity and Ethics		5	8, 13, 14, 15, 19	0.85	
Giordano-Mulligan [22,23]	Personal Integrity	Self-Awareness	6	1, 2, 3, 4, 5, 6	0.90	0.99
		Moral Ethical Courage	4	7, 8, 9, 10	0.89	
	Transparency	Relational Integrity	7	11, 12, 13, 14, 15, 16, 17	0.96	
		Shared Decision Making	6	18, 19, 20, 21, 22, 23	0.95	
	Altruism	Caring	6	24, 25, 26, 27, 28, 29	0.97	
Wang et al. [37]	Self-Awareness		6	1, 2, 3, 4, 5, 6	0.96	0.97
	Moral Ethical Courage		4	7, 8, 9, 10	0.96	
	Relational Integrity		7	11, 12, 13, 14, 15, 16, 17	0.97	
	Shared Decision Making		6	18, 19, 20, 21, 22, 23	0.92	
	Caring		6	24, 25, 26, 27, 28, 29	0.94	

## Data Availability

Restrictions apply to the availability of these data. Data were obtained from a third party and are available with the permission of the third party.

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
