# Peer review of "Translation, Cross-Cultural Adaptation, and Psychometric Validation of the Authentic Nurse Leadership Questionnaire for the Portuguese Context: A Methodological Study"

_nursrep, 2025, doi:10.3390/nursrep15100362_

Round 1
Reviewer 1 Report
Comments and Suggestions for Authors
Dear authors,
Adapting and validating a questionnaire that analyzes leadership in nursing is always an interesting topic due to its significant implications for the quality of healthcare.
The methodology should explain the psychometric characteristics of the original questionnaire to be validated: statistical analysis of the items, reliability (internal consistency, homogeneity, test-retest reliability), and validity (logical, content, construct).
The specific modifications, if any, made after the cross-cultural adaptation of the questionnaire to Portuguese should also be indicated.
I believe that a statistical analysis of the items should have been performed, primarily assessing the standard deviation and symmetry, to verify their discriminatory capacity and representativeness in this population. These values should be reflected in a table in the Results section.
Internal consistency assesses the correlation between items; to assess homogeneity, it is necessary to analyze the item-total correlation and the corrected item-total correlation, which does not appear in the study. This correlation should be performed and reflected in a table in the Results section.
What were the test-retest reliability results? They are not indicated in the Results section. If the dimensions assessed by this questionnaire vary, a 14-day period may be excessive, as it allows factors external to the questionnaire to modify the results. If this is the case, this should be indicated in the Limitations section.
After performing the Bartlett and KMO tests, a Principal Components Analysis or a Common Factor Analysis with rotation should be performed to obtain the factors and items corresponding to each. The type of analysis performed should be indicated.
I believe a table explaining the total variance explained and another with the rotated component matrix should be created, similar to those generated by SPSS.
The acronyms for the different goodness-of-fit indices used should be explained.
A table should also be created comparing the two theoretical models analyzed (5-dimensional and 3-dimensional) with their respective goodness-of-fit indices.
In my opinion, tables of the results obtained are missing. Such tables would facilitate readers' understanding.
The exploratory factor analysis reduced the five factors in the original questionnaire and the Chinese adaptation to three. I believe the possible implications of these results should be indicated in the discussion and the sociocultural characteristics of the Portuguese population that explain them should be specified.
I believe the limitations derived from the type of sampling used should be included in the Limitations section rather than in the Conclusions section.
The bibliographic references are adequate and mostly up-to-date.
Kind regards.
Reviewer 2 Report
Comments and Suggestions for Authors
Line 93 states that the population (the total number of participants who could be included in the study) is 510, but there are no sources to verify this data. Therefore, the sample size could not be justified as sufficient or insufficient (line 98).
Reference about the sources of the instrument (lines 101 and 102) could be omited.
Lines 123 to 125 describe the comparison and discussion of the two previous versions of the instrument, but there is no quantitative data, such as Cohen's kappa or Krippendorff's alpha, to support the conclusions.
The back-translation procedure is described from lines 126 to 131, but no references are given for the results.
Lines 138 to 141 describe the coincidence across versions T1 and T2, where 17 items coincided. However, there is no reference to how the inconsistencies among the remaining items were treated or resolved.
Lines 182-185 could be shortened.
Lines 188-189 are inconsistent with previous data (lines 186-187). I suggest reviewing and adjusting them.
The reliability results were presented using Cronbach's alpha, but it is better to use McDonald's omega because it aligns with Likert scales. Cronbach's alpha was designed for quantitative scales. Additionally, it is recommended that the average inter-item correlation be calculated to identify and prevent systematic errors (items that could measure similar topics). This correlation should be below 0.6. This is also recommended because Cronbach's alphas are too high, which could indicate redundancy among the items.
The test-retest procedures were developed too close in time. It is better to have a broader timeframe to minimize recency effects (this could be discussed as a procedure limitation).
It is better to include the Bartlett's test of sphericity value rather than just the p-value.
It is better to verify the chosen rotation based on the initial correlation among the factors (those above 0.3 must be orthogonal).
Line 244 must be updated because the minimum data reported in lines 211 to 216 is above 0.85, according to Cronbach's Alpha report.
Line 267 describes a reorganization of the items, but it would be better to clarify the implications.
The explained variance in Table 1 must be corrected according to the data described in lines 248 to 262, as well as Cronbach's alpha according to lines 211 to 215.
Line 278 refers to the use of the Mahalanobis distance (D²), but there is no information about the values or uses of this statistical tool.
It is essential to make adjustments to Figure 1 to ensure optimal readability.

Reviewer 3 Report
Comments and Suggestions for Authors
Dear authors,
Working on the adaptation and validation of instruments that measure leadership in nursing is particularly valuable given the significant healthcare relevance of this professional competency.
The title provides adequate information about the topic of the article and would facilitate its search in databases.
The abstract is well-structured and adequately reports on the most relevant aspects of the study.
The introduction is concise, although I believe it addresses the main aspects related to leadership and the Portuguese validation of the study questionnaire.
The objective is clear and well-defined.
The methodology explains the adaptation and validation process of the questionnaire in detail, allowing for replication of the study. To be more comprehensive, it should analyze other aspects such as the standard deviation and skewness of the items, as well as the item-total and corrected item-total coefficients. It should also indicate what type of test was used to obtain the factors in the exploratory factor analysis.
The results are detailed but are presented primarily in written form and in a single table. More tables should be included to improve the aesthetics of the document and allow those interested in this topic to draw their own conclusions.
The discussion is extensive and adequately analyzes the results obtained, comparing them with the existing literature, although aspects related to the reduction in factors compared to the original questionnaire should also be addressed.
The analysis of the study's limitations is detailed, although I believe that those derived from the type of sampling used should also be included in this section, along with the conclusions.
The conclusions are consistent with the results obtained and respond to the proposed objective.
The references are abundant and appropriate to the topic of study.
Kind regards.
Round 2
Reviewer 1 Report
Comments and Suggestions for Authors
Dear authors,
I have no further comments on your manuscript.
Kind regards.
Reviewer 2 Report
Comments and Suggestions for Authors
Thanks for taking ino account my suggestion.
Still persist minimal corrections to do:
Table 2: is Items (no itens)
Figure 1: do your best to make ir readeble.
Reviewer 3 Report
Comments and Suggestions for Authors
Dear Authors,
I consider that the manuscript has been sufficiently improved and I have no additional comments or suggestions.
Kind regards.
